# Optimization Monte Carlo: Efficient and Embarrassingly Parallel Likelihood-Free Inference

**Edward Meeds**
Informatics Institute
University of Amsterdam
tmeeds@gmail.com

**Max Welling**\*
Informatics Institute
University of Amsterdam
welling.max@gmail.com

## Abstract

We describe an embarrassingly parallel, anytime Monte Carlo method for likelihood-free models. The algorithm starts with the view that the stochasticity of the pseudo-samples generated by the simulator can be controlled externally by a vector of random numbers $u$, in such a way that the outcome, knowing $u$, is deterministic. For each instantiation of $u$ we run an optimization procedure to minimize the distance between summary statistics of the simulator and the data. After reweighing these samples using the prior and the Jacobian (accounting for the change of volume in transforming from the space of summary statistics to the space of parameters) we show that this weighted ensemble represents a Monte Carlo estimate of the posterior distribution. The procedure can be run embarrassingly parallel (each node handling one sample) and anytime (by allocating resources to the worst performing sample). The procedure is validated on six experiments.

## 1 Introduction

Computationally demanding simulators are used across the full spectrum of scientific and industrial applications, whether one studies embryonic morphogenesis in biology, tumor growth in cancer research, colliding galaxies in astronomy, weather forecasting in meteorology, climate changes in the environmental science, earthquakes in seismology, market movement in economics, turbulence in physics, brain functioning in neuroscience, or fabrication processes in industry. Approximate Bayesian computation (ABC) forms a large class algorithms that aims to sample from the posterior distribution over parameters for these likelihood-free (a.k.a. simulator based) models. Likelihood-free inference, however, is notoriously inefficient in terms of the number of simulation calls per independent sample. Further, like regular Bayesian inference algorithms, care must be taken so that posterior sampling targets the correct distribution.

The simplest ABC algorithm, ABC rejection sampling, can be fully parallelized by running independent processes with no communication or synchronization requirements. I.e. it is an *embarrassingly parallel* algorithm. Unfortunately, as the most inefficient ABC algorithm, the benefits of this title are limited. There has been considerable progress in distributed MCMC algorithms aimed at large-scale data problems [2, 1]. Recently, a sequential Monte Carlo (SMC) algorithm called "the particle cascade" was introduced that emits streams of samples asynchronously with minimal memory management and communication [17]. In this paper we present an alternative embarrassingly parallel sampling approach: each processor works independently, at full capacity, and will indefinitely emit independent samples. The main trick is to pull random number generation outside of the simulator and treat the simulator as a deterministic piece of code. We then minimize the difference

between observations and the simulator output over its input parameters and weight the final (optimized) parameter value with the prior and the (inverse of the) Jacobian. We show that the resulting weighted ensemble represents a Monte Carlo estimate of the posterior. Moreover, we argue that the error of this procedure is $\mathcal{O}(\epsilon)$ if the optimization gets $\epsilon$-close to the optimal value. This "Optimization Monte Carlo" (OMC) has several advantages: 1) it can be run embarrassingly parallel, 2) the procedure generates independent samples and 3) the core procedure is now optimization rather than MCMC. Indeed, optimization as part of a likelihood-free inference procedure has recently been proposed [12]; using a probabilistic model of the mapping from parameters to differences between observations and simulator outputs, they apply "Bayesian Optimization" (e.g. [13, 21]) to efficiently perform posterior inference. Note also that since random numbers have been separated out from the simulator, powerful tools such as "automatic differentiation" (e.g. [14]) are within reach to assist with the optimization. In practice we find that OMC uses far fewer simulations per sample than alternative ABC algorithms.

The approach of controlling randomness as part of an inference procedure is also found in a related class of parameter estimation algorithms called *indirect inference* [11]. Connections between ABC and indirect inference have been made previously by [7] as a novel way of creating summary statistics. An indirect inference perspective led to an independently developed version of OMC called the "reverse sampler" [9, 10].

In Section 2 we briefly introduce ABC and present it from a novel viewpoint in terms of random numbers. In Section 3 we derive ABC through optimization from a geometric point of view, then proceed to generalize it to higher dimensions. We show in Section 4 extensive evidence of the correctness and efficiency of our approach. In Section 5 we describe the outlook for optimization-based ABC.

## 2 ABC Sampling Algorithms

The primary interest in ABC is the posterior of simulator parameters $\boldsymbol{\theta}$ given a vector of (statistics of) observations $\mathbf{y}$, $p(\boldsymbol{\theta}|\mathbf{y})$. The likelihood $p(\mathbf{y}|\boldsymbol{\theta})$ is generally not available in ABC. Instead we can use the simulator as a generator of pseudo-samples $\mathbf{x}$ that reside in the same space as $\mathbf{y}$. By treating $\mathbf{x}$ as auxiliary variables, we can continue with the Bayesian treatment:

$$p(\boldsymbol{\theta}|\mathbf{y}) = \frac{p(\boldsymbol{\theta})p(\mathbf{y}|\boldsymbol{\theta})}{p(\mathbf{y})} \approx \frac{p(\boldsymbol{\theta})\int p_{\epsilon}(\mathbf{y}|\mathbf{x})p(\mathbf{x}|\boldsymbol{\theta})\,d\mathbf{x}}{\int p(\boldsymbol{\theta})\int p_{\epsilon}(\mathbf{y}|\mathbf{x})p(\mathbf{x}|\boldsymbol{\theta})\,d\mathbf{x}\,d\boldsymbol{\theta}} \tag{1}$$

Of particular importance is the choice of *kernel* measuring the discrepancy between observations $\mathbf{y}$ and pseudo-data $\mathbf{x}$. Popular choices for kernels are the Gaussian kernel and the uniform $\epsilon$-tube/ball. The bandwidth parameter $\epsilon$ (which may be a vector $\boldsymbol{\epsilon}$ accounting for relative importance of each statistic) plays critical role: small $\epsilon$ produces more accurate posteriors, but is more computationally demanding, whereas large $\epsilon$ induces larger error but is cheaper.

We focus our attention on population-based ABC samplers, which include rejection sampling, importance sampling (IS), sequential Monte Carlo (SMC) [6, 20] and population Monte Carlo [3]. In rejection sampling, we draw parameters from the prior $\boldsymbol{\theta} \sim p(\boldsymbol{\theta})$, then run a simulation at those parameters $\mathbf{x} \sim p(\mathbf{x}|\boldsymbol{\theta})$; if the discrepancy $\rho(\mathbf{x}, \mathbf{y}) < \epsilon$, then the particle is accepted, otherwise it is rejected. This is repeated until $n$ particles are accepted. Importance sampling generalizes rejection sampling using a proposal distribution $q_{\phi}(\boldsymbol{\theta})$ instead of the prior, and produces samples with weights $w_i \propto p(\boldsymbol{\theta})/q(\boldsymbol{\theta})$. SMC extends IS to multiple rounds with decreasing $\epsilon$, adapting their particles after each round, such that each new population improves the approximation to the posterior. Our algorithm has similar qualities to SMC since we generate a population of $n$ weighted particles, but differs significantly since our particles are produced by independent optimization procedures, making it completely parallel.

## 3 A Parallel and Efficient ABC Sampling Algorithm

Inherent in our assumptions about the simulator is that internally there are calls to a random number generator which produces the stochasticity of the pseudo-samples. We will assume for the moment that this can be represented by a vector of uniform random numbers $\boldsymbol{u}$ which, if known, would make the simulator deterministic. More concretely, we assume that any simulation output $\mathbf{x}$ can be represented as a deterministic function of parameters $\boldsymbol{\theta}$ and a vector of random numbers $\boldsymbol{u}$,

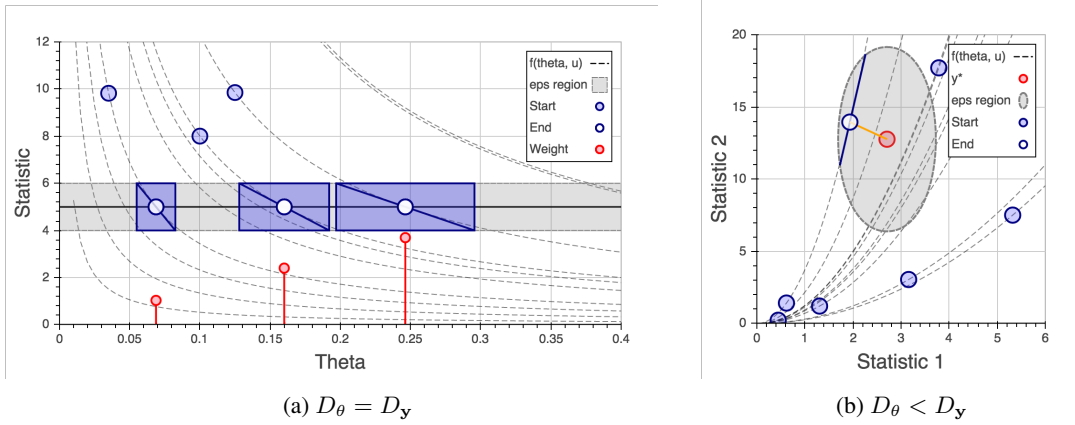

(a) $D_\theta = D_\mathbf{y}$          (b) $D_\theta < D_\mathbf{y}$

Figure 1: Illustration of OMC geometry. (a) Dashed lines indicate contours $f(\theta, \boldsymbol{u})$ over $\theta$ for several $\boldsymbol{u}$. For three values of $\boldsymbol{u}$, their initial and optimal $\theta$ positions are shown (solid blue/white circles). Within the grey acceptance region, the Jacobian, indicated by the blue diagonal line, describes the relative change in volume induced in $f(\theta, \boldsymbol{u})$ from a small change in $\theta$. Corresponding weights $\propto 1/|J|$ are shown as vertical stems. (b) When $D_\theta < D_\mathbf{y}$, here $1 < 2$, the change in volume is proportional to the length of the line segment inside the ellipsoid ($|\mathbf{J}^T\mathbf{J}|^{1/2}$). The orange line indicates the projection of the observation onto the contour of $f(\theta, \boldsymbol{u})$ (in this case, identical to the optimal).

i.e. $\mathbf{x} = \boldsymbol{f}(\boldsymbol{\theta}, \boldsymbol{u})$. This assumption has been used previously in ABC, first in "coupled ABC" [16] and also in an application of Hamiltonian dynamics to ABC [15]. We do not make any further assumptions regarding $\boldsymbol{u}$ or $p(\boldsymbol{u})$, though for some problems their dimension and distribution may be known a priori. In these cases it may be worth employing Sobol or other low-discrepancy sequences to further improve the accuracy of any Monte Carlo estimates.

We will first derive a dual representation for the ABC likelihood function $p_\epsilon(\mathbf{y}|\boldsymbol{\theta})$ (see also [16]),

$$p_\epsilon(\mathbf{y}|\boldsymbol{\theta}) = \int p_\epsilon(\mathbf{y}|\mathbf{x})p(\mathbf{x}|\boldsymbol{\theta})\,d\mathbf{x} = \int\int p_\epsilon(\mathbf{y}|\mathbf{x})\mathbb{I}[\mathbf{x} = \boldsymbol{f}(\boldsymbol{\theta}, \boldsymbol{u})]p(\boldsymbol{u})\,d\mathbf{x}d\boldsymbol{u} \tag{2}$$

$$= \int p_\epsilon(\mathbf{y}|\boldsymbol{f}(\boldsymbol{\theta}, \boldsymbol{u}))p(\boldsymbol{u})\,d\boldsymbol{u} \tag{3}$$

leading to the following Monte Carlo approximation of the ABC posterior,

$$p_\epsilon(\boldsymbol{\theta}|\mathbf{y}) \propto p(\boldsymbol{\theta})\int p(\boldsymbol{u})p_\epsilon(\mathbf{y}|\boldsymbol{f}(\boldsymbol{u}, \boldsymbol{\theta}))\,d\boldsymbol{u} \approx \frac{1}{n}\sum_i p_\epsilon(\mathbf{y}|\boldsymbol{f}(\boldsymbol{u}_i, \boldsymbol{\theta}))p(\boldsymbol{\theta}) \quad \boldsymbol{u}_i \sim p(\boldsymbol{u}) \tag{4}$$

Since $p_\epsilon$ is a kernel that only accepts arguments $\mathbf{y}$ and $\boldsymbol{f}(\boldsymbol{u}_i, \boldsymbol{\theta})$ that are $\epsilon$ close to each other (for values of $\epsilon$ that are as small as possible), Equation 4 tells us that we should first sample values for $\boldsymbol{u}$ from $p(\boldsymbol{u})$ and then for each such sample find the value for $\boldsymbol{\theta}_i^o$ that results in $\mathbf{y} = \boldsymbol{f}(\boldsymbol{\theta}_i^o, \boldsymbol{u})$. In practice we want to drive these values as close to each other as possible through optimization and accept an $\mathcal{O}(\epsilon)$ error if the remaining distance is still $\mathcal{O}(\epsilon)$. Note that apart from sampling the values for $\boldsymbol{u}$ this procedure is deterministic and can be executed completely in parallel, i.e. without any communication. In the following we will assume a single observation vector $\mathbf{y}$, but the approach is equally applicable to a dataset of $N$ cases.

### 3.1 The case $D_\theta = D_\mathbf{y}$

We will first study the case when the number of parameters $\boldsymbol{\theta}$ is equal to the number of summary statistics $\mathbf{y}$. To understand the derivation it helps to look at Figure 1a which illustrates the derivation for the one dimensional case. In the following we use the following abbreviation: $f_i(\boldsymbol{\theta})$ stands for $f(\boldsymbol{\theta}, \boldsymbol{u}_i)$. The general idea is that we want to write the approximation to the posterior as a mixture of small uniform balls (or delta peaks in the limit):

$$p(\boldsymbol{\theta}|\mathbf{y}) \approx \frac{1}{n}\sum_i p_\epsilon(\mathbf{y}|\boldsymbol{f}(\boldsymbol{u}_i, \boldsymbol{\theta}))p(\boldsymbol{\theta}) \approx \frac{1}{n}\sum_i w_i U_\epsilon(\boldsymbol{\theta}|\boldsymbol{\theta}_i^*)p(\boldsymbol{\theta}) \tag{5}$$

with $w_i$ some weights that we will derive shortly. Then, if we make $\epsilon$ small enough we can replace any average of a sufficiently smooth function $h(\boldsymbol{\theta})$ w.r.t. this approximate posterior simply by evaluating $h(\boldsymbol{\theta})$ at some arbitrarily chosen points inside these balls (for instance we can take the center of the ball $\boldsymbol{\theta}_i^*$),

$$\int h(\boldsymbol{\theta})p(\boldsymbol{\theta}|\mathbf{y})\,d\boldsymbol{\theta} \approx \frac{1}{n}\sum_i h(\boldsymbol{\theta}_i^*)w_i p(\boldsymbol{\theta}_i^*) \tag{6}$$

To derive this expression we first assume that:

$$p_{\boldsymbol{\epsilon}}(\mathbf{y}|\boldsymbol{f}_i(\boldsymbol{\theta})) = C(\boldsymbol{\epsilon})\mathbb{I}[||\mathbf{y} - \boldsymbol{f}_i(\boldsymbol{\theta})||^2 \leq \epsilon^2] \tag{7}$$

i.e. a ball of radius $\epsilon$. $C(\boldsymbol{\epsilon})$ is the normalizer which is immaterial because it cancels in the posterior. For small enough $\epsilon$ we claim that we can linearize $\boldsymbol{f}_i(\boldsymbol{\theta})$ around $\boldsymbol{\theta}_i^o$:

$$\hat{\boldsymbol{f}}_i(\boldsymbol{\theta}) = \boldsymbol{f}_i(\boldsymbol{\theta}_i^o) + \mathbf{J}_i^o(\boldsymbol{\theta} - \boldsymbol{\theta}_i^o) + R_i, \qquad R_i = \mathcal{O}(||\boldsymbol{\theta} - \boldsymbol{\theta}_i^o||^2) \tag{8}$$

where $\mathbf{J}_i^o$ is the Jacobian matrix with columns $\frac{\partial \boldsymbol{f}_i(\boldsymbol{\theta}_i^o)}{\partial \theta_d}$. We take $\boldsymbol{\theta}_i^o$ to be the end result of our optimization procedure for sample $\boldsymbol{u}_i$. Using this we thus get,

$$||\mathbf{y} - \boldsymbol{f}_i(\boldsymbol{\theta})||^2 \approx ||(\mathbf{y} - \boldsymbol{f}_i(\boldsymbol{\theta}_i^o)) - \mathbf{J}_i^o(\boldsymbol{\theta} - \boldsymbol{\theta}_i^o) - R_i||^2 \tag{9}$$

We first note that since we assume that our optimization has ended up somewhere inside the ball defined by $||\mathbf{y} - \boldsymbol{f}_i(\boldsymbol{\theta})||^2 \leq \epsilon^2$ we can assume that $||\mathbf{y} - \boldsymbol{f}_i(\boldsymbol{\theta}_i^o)|| = \mathcal{O}(\epsilon)$. Also, since we only consider values for $\boldsymbol{\theta}$ that satisfy $||\mathbf{y} - \boldsymbol{f}_i(\boldsymbol{\theta})||^2 \leq \epsilon^2$, and furthermore assume that the function $\boldsymbol{f}_i(\boldsymbol{\theta})$ is Lipschitz continuous in $\boldsymbol{\theta}$ it follows that $||\boldsymbol{\theta} - \boldsymbol{\theta}_i^o|| = \mathcal{O}(\epsilon)$ as well. All of this implies that we can safely ignore the remaining term $R_i$ (which is of order $\mathcal{O}(||\boldsymbol{\theta} - \boldsymbol{\theta}_i^o||^2) = \mathcal{O}(\epsilon^2)$) if we restrict ourselves to the volume inside the ball.

The next step is to view the term $\mathbb{I}[||\mathbf{y} - \boldsymbol{f}_i(\boldsymbol{\theta})||^2 \leq \epsilon^2]$ as a distribution in $\boldsymbol{\theta}$. With the Taylor expansion this results in,

$$\mathbb{I}[(\boldsymbol{\theta} - \boldsymbol{\theta}_i^o - \mathbf{J}_i^{o,-1}(\mathbf{y} - \boldsymbol{f}_i(\boldsymbol{\theta}_i^o)))^T \mathbf{J}_i^{oT}\mathbf{J}_i^o(\boldsymbol{\theta} - \boldsymbol{\theta}_i^o - \mathbf{J}_i^{o,-1}(\mathbf{y} - \boldsymbol{f}_i(\boldsymbol{\theta}_i^o))) \leq \epsilon^2] \tag{10}$$

This represents an ellipse in $\boldsymbol{\theta}$-space with a centroid $\boldsymbol{\theta}_i^*$ and volume $V_i$ given by

$$\boldsymbol{\theta}_i^* = \boldsymbol{\theta}_i^o + \mathbf{J}_i^{o,-1}(\mathbf{y} - \boldsymbol{f}_i(\boldsymbol{\theta}_i^o)) \qquad V_i = \frac{\gamma}{\sqrt{\det(\mathbf{J}_i^{oT}\mathbf{J}_i^o)}} \tag{11}$$

with $\gamma$ a constant independent of $i$. We can approximate the posterior now as,

$$p(\boldsymbol{\theta}|\mathbf{y}) \approx \frac{1}{\kappa}\sum_i \frac{U_{\boldsymbol{\epsilon}}(\boldsymbol{\theta}|\boldsymbol{\theta}_i^*)p(\boldsymbol{\theta})}{\sqrt{\det(\mathbf{J}_i^{oT}\mathbf{J}_i^o)}} \approx \frac{1}{\kappa}\sum_i \frac{\delta(\boldsymbol{\theta} - \boldsymbol{\theta}_i^*)p(\boldsymbol{\theta}_i^*)}{\sqrt{\det(\mathbf{J}_i^{oT}\mathbf{J}_i^o)}} \tag{12}$$

where in the last step we have send $\epsilon \to 0$. Finally, we can compute the constant $\kappa$ through normalization, $\kappa = \sum_i p(\boldsymbol{\theta}_i^*)\det(\mathbf{J}_i^{oT}\mathbf{J}_i^o)^{-1/2}$. The whole procedure is accurate up to errors of the order $\mathcal{O}(\epsilon^2)$, and it is assumed that the optimization procedure delivers a solution that is located within the epsilon ball. If one of the optimizations for a certain sample $\boldsymbol{u}_i$ did not end up within the epsilon ball there can be two reasons: 1) the optimization did not converge to the optimal value for $\boldsymbol{\theta}$, or 2) for this value of $\boldsymbol{u}$ there is no solution for which $\boldsymbol{f}(\boldsymbol{\theta}|\boldsymbol{u})$ can get within a distance $\epsilon$ from the observation $\mathbf{y}$. If we interpret $\epsilon$ as our uncertainty in the observation $\mathbf{y}$, and we assume that our optimization succeeded in finding the best possible value for $\boldsymbol{\theta}$, then we should simply reject this sample $\boldsymbol{\theta}_i$. However, it is hard to detect if our optimization succeeded and we may therefore sometimes reject samples that should not have been rejected. Thus, one should be careful not to create a bias against samples $\boldsymbol{u}_i$ for which the optimization is difficult. This situation is similar to a sampler that will not mix to remote local optima in the posterior distribution.

## 3.2 The case $D_\theta < D_{\mathbf{y}}$

This is the overdetermined case and here the situation as depicted in Figure 1b is typical: the manifold that $\boldsymbol{f}(\boldsymbol{\theta}, \boldsymbol{u}_i)$ traces out as we vary $\boldsymbol{\theta}$ forms a lower dimensional surface in the $D_{\mathbf{y}}$ dimensional enveloping space. This manifold may or may not intersect with the sphere centered at the observation $\mathbf{y}$ (or ellipsoid, for the general case $\boldsymbol{\epsilon}$ instead of $\epsilon$). Assume that the manifold does intersect the

epsilon ball but not $\mathbf{y}$. Since we trust our observation up to distance $\epsilon$, we may simple choose to pick the closest point $\boldsymbol{\theta}_i^*$ to $\mathbf{y}$ on the manifold, which is given by,

$$\boldsymbol{\theta}_i^* = \boldsymbol{\theta}_i^o + \mathbf{J}_i^{o\dagger}(\mathbf{y} - \boldsymbol{f}_i(\boldsymbol{\theta}_i^o)) \qquad \mathbf{J}_i^{o\dagger} = (\mathbf{J}_i^{oT}\mathbf{J}_i^o)^{-1}\mathbf{J}_i^{oT} \tag{13}$$

where $\mathbf{J}_i^{o\dagger}$ is the pseudo-inverse. We can now define our ellipse around this point, shifting the center of the ball from $\mathbf{y}$ to $\boldsymbol{f}_i(\boldsymbol{\theta}_i^*)$ (which do not coincide in this case). The uniform distribution on the ellipse in $\boldsymbol{\theta}$-space is now defined in the $D_\theta$ dimensional manifold and has volume $V_i = \gamma \det(\mathbf{J}_i^{oT}\mathbf{J}_i^o)^{-1/2}$. So once again we arrive at almost the same equation as before (Eq. 12) but with the slightly different definition of the point $\theta_i^*$ given by Eq. 13. Crucially, since $||\mathbf{y} - \boldsymbol{f}_i(\boldsymbol{\theta}_i^*)|| \leq \epsilon^2$ and if we assume that our optimization succeeded, we will only make mistakes of order $\mathcal{O}(\epsilon^2)$.

### 3.3 The case $D_\theta > D_\mathbf{y}$

This is the underdetermined case in which it is typical that entire manifolds (e.g. hyperplanes) may be a solution to $||\mathbf{y} - \boldsymbol{f}_i(\boldsymbol{\theta}_i^*)|| = 0$. In this case we can not approximate the posterior with a mixture of point masses and thus the procedure does not apply. However, the case $D_\theta > D_\mathbf{y}$ is less interesting than the other ones above as we expect to have more summary statistics than parameters for most problems.

## 4 Experiments

The goal of these experiments is to demonstrate 1) the correctness of OMC and 2) the relative efficiency of OMC in relation to two sequential MC algorithms, SMC (aka population MC [3]) and adaptive weighted SMC [5]. To demonstrate correctness, we show histograms of weighted samples along with the true posterior (when known) and, for three experiments, the exact OMC weighted samples (when the exact Jacobian and optimal $\theta$ is known). To demonstrate efficiency, we compute the mean simulations per sample (SS)—the number of simulations required to reach an $\epsilon$ threshold— and the effective sample size (ESS), defined as $1/\mathbf{w}^T\mathbf{w}$. Additionally, we may measure ESS/n, the fraction of effective samples in the population. ESS is a good way of detecting whether the posterior is dominated by a few particles and/or how many particles achieve discrepancy less than epsilon.

There are several algorithmic options for OMC. The most obvious is to spawn independent processes, draw $\boldsymbol{u}$ for each, and optimize until $\epsilon$ is reached (or a max nbr of simulations run), then compute Jacobians and particle weights. Variations could include keeping a sorted list of discrepancies and allocating computational resources to the worst particle. However, to compare OMC with SMC, in this paper we use a sequential version of OMC that mimics the epsilon rounds of SMC. Each simulator uses different optimization procedures, including Newton's method for smooth simulators, and random walk optimization for others; Jacobians were computed using one-sided finite differences. To limit computational expense we placed a max of 1000 simulations per sample per round for all algorithms. Unless otherwise noted, we used $n = 5000$ and repeated runs 5 times; lack of error bars indicate very low deviations across runs. We also break some of the notational convention used thus far so that we can specify exactly how the random numbers translate into pseudo-data and the pseudo-data into statistics. This is clarified for each example. Results are explained in Figures 2 to 4.

### 4.1 Normal with Unknown Mean and Known Variance

The simplest example is the inference of the mean $\theta$ of a univariate normal distribution with known variance $\sigma^2$. The prior distribution $\pi(\theta)$ is normal with mean $\theta_0$ and variance $k\sigma^2$, where $k > 0$ is a factor relating the dispersions of $\theta$ and the data $y_n$. The simulator can generate data according to the normal distribution, or deterministically if the random effects $r_{u_m}$ are known:

$$\pi(x_m|\theta) = \mathcal{N}(x_m|\theta, \sigma^2) \quad \implies \quad x_m = \theta + r_{u_m} \tag{14}$$

where $r_{u_m} = \sigma\sqrt{2}\,\mathrm{erf}^{-1}(2u_m - 1)$ (using the inverse CDF). A sufficient statistic for this problem is the average $s(\mathbf{x}) = \frac{1}{M}\sum_m x_m$. Therefore we have $f(\theta, \boldsymbol{u}) = \theta + R(\boldsymbol{u})$ where $R(\boldsymbol{u}) = \sum r_{u_m}/M$ (the average of the random effects). In our experiment we set $M = 2$ and $y = 0$. The *exact* Jacobian and $\theta_i^o$ can be computed for this problem: for a draw $\boldsymbol{u}_i$, $J_i = 1$; if $s(\mathbf{y})$ is the mean of the observations $\mathbf{y}$, then by setting $f(\theta_i^o, \boldsymbol{u}_i) = s(\mathbf{y})$ we find $\theta_i^o = s(\mathbf{y}) - R(\boldsymbol{u}_i)$. Therefore the exact weights are $w_i \propto \pi(\theta_i^o)$, allowing us to compare directly with an exact posterior based on our dual representation by $\boldsymbol{u}$ (shown by orange circles in Figure 2 top-left). We used Newton's method to optimize each particle.

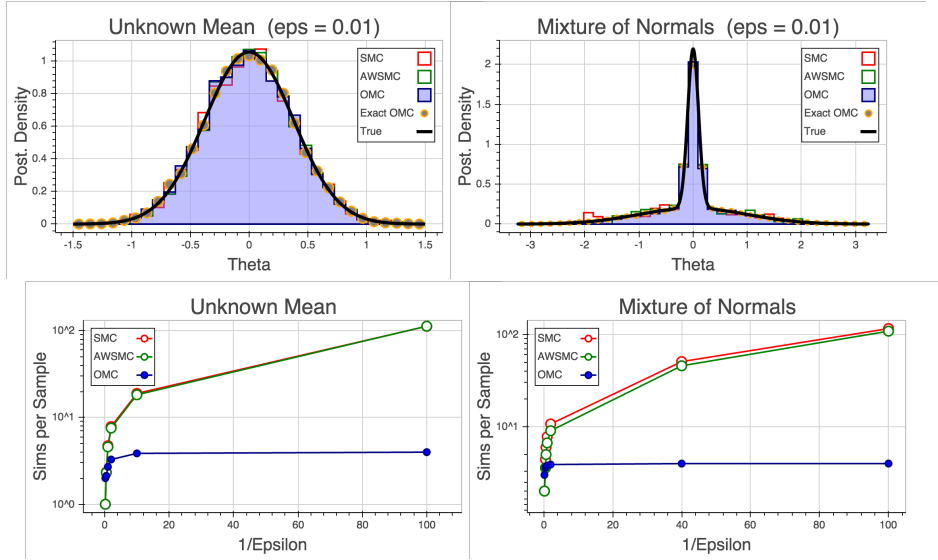

Figure 2: **Left:** Inference of unknown mean. For $\epsilon$ 0.1, OMC uses 3.7 SS; AW/SMC uses 20/20 SS; at $\epsilon$ 0.01, OMC uses 4 SS (only 0.3 SS more), and SMC jumps to 110 SS. For all algorithms and $\epsilon$ values, ESS/n=1. **Right:** Inference for mixture of normals. Similar results for OMC; at $\epsilon$ 0.025 AW/SMC had $40/50$ SS and at $\epsilon$ 0.01 has $105/120$ SS. The ESS/n remained at 1 for OMC, but decreased to $0.06/0.47$ (AW/SMC) at $\epsilon$ 0.025, and 0.35 for both at $\epsilon$ 0.01. Not only does the ESS remain high for OMC, but it also represents the tails of the distribution well, even at low $\epsilon$.

## 4.2 Normal Mixture

A standard illustrative ABC problem is the inference of the mean $\theta$ of a mixture of two normals [19, 3, 5]: $p(x|\theta) = \rho\,\mathcal{N}(\theta, \sigma_1^2) + (1-\rho)\,\mathcal{N}(\theta, \sigma_2^2)$, with $\pi(\theta) = \mathcal{U}(\theta_a, \theta_b)$ where hyperparameters are $\rho = 1/2$, $\sigma_1^2 = 1$, $\sigma_2^2 = 1/100$, $\theta_a = -10$, $\theta_b = 10$, and a single observation scalar $y = 0$. For this problem $M = 1$ so we drop the subscript $m$. The true posterior is simply $p(\theta|y = 0) \propto \rho\,\mathcal{N}(\theta, \sigma_1^2) + (1-\rho)\,\mathcal{N}(\theta, \sigma_2^2)$, $\theta \in \{-10, 10\}$. In this problem there are two random numbers $u_1$ and $u_2$, one for selecting the mixture component and the other for the random innovation; further, the statistic is the identity, i.e. $s(x) = x$:

$$x = [u_1 < \rho](\theta + \sigma_1\sqrt{2}\,\mathrm{erf}(2u_2 - 1)) + [u_1 \geq \rho](\theta + \sigma_2\sqrt{2}\,\mathrm{erf}(2u_2 - 1)) \quad (15)$$

$$= \theta + \sqrt{2}\,\mathrm{erf}(2u_2 - 1)\sigma_1^{[u_1<\rho]}\sigma_2^{[u_1\geq\rho]} = \theta + R(\boldsymbol{u}) \quad (16)$$

where $R(\boldsymbol{u}) = \sqrt{2}\,\mathrm{erf}(2u_2 - 1)\sigma_1^{[u_1<\rho]}\sigma_2^{[u_1\geq\rho]}$. As with the previous example, the Jacobian is 1 and $\theta_i^o = y - R(\boldsymbol{u}_i)$ is known exactly. This problem is notable for causing performance issues in ABC-MCMC [19] and its difficulty in targeting the tails of the posterior [3]; this is not the case for OMC.

## 4.3 Exponential with Unknown Rate

In this example, the goal is to infer the rate $\theta$ of an exponential distribution, with a gamma prior $p(\theta) = \mathrm{Gamma}(\theta|\alpha, \beta)$, based on $M$ draws from $\mathrm{Exp}(\theta)$:

$$p(x_m|\theta) = \mathrm{Exp}(x_m|\theta) = \theta\exp(-\theta x_m) \quad \Longrightarrow \quad x_m = -\frac{1}{\theta}\ln(1 - u_m) = \frac{1}{\theta}r_{u_m} \quad (17)$$

where $r_{u_m} = -\ln(1 - u_m)$ (the inverse CDF of the exponential). A sufficient statistic for this problem is the average $s(\mathbf{x}) = \sum_m x_m/M$. Again, we have exact expressions for the Jacobian and $\theta_i^o$, using $f(\theta, \boldsymbol{u}_i) = R(\boldsymbol{u}_i)/\theta$, $J_i = -R(\boldsymbol{u}_i)/\theta^2$ and $\theta_i^o = R(\boldsymbol{u}_i)/s(\mathbf{y})$. We used $M = 2$, $s(\mathbf{y}) = 10$ in our experiments.

## 4.4 Linked Mean and Variance of Normal

In this example we link together the mean and variance of the data generating function as follows:

$$p(x_m|\theta) = \mathcal{N}(x_m|\theta, \theta^2) \quad \Longrightarrow \quad x_m = \theta + \theta\sqrt{2}\,\mathrm{erf}^{-1}(2u_m - 1) = \theta r_{u_m} \quad (18)$$

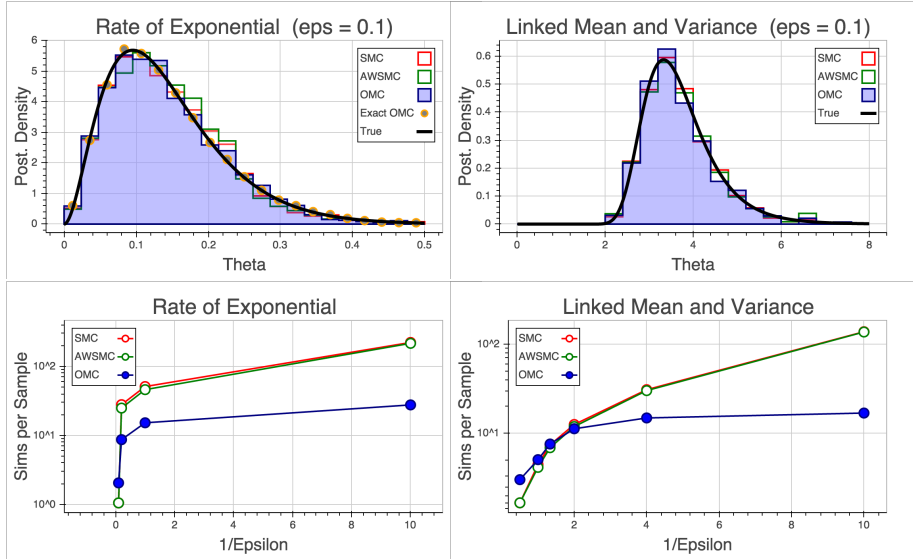

Figure 3: **Left:** Inference of rate of exponential. A similar result wrt SS occurs for this experiment: at $\epsilon$ 1, OMC had 15 v 45/50 for AW/SMC; at $\epsilon$ 0.01, SS was 28 OMC v 220 AW/SMC. ESS/n dropping with below 1: OMC drops at $\epsilon$ 1 to 0.71 v 0.97 for SMC; at $\epsilon$ 0.1 ESS/n remains the same. **Right:** Inference of linked normal. ESS/n drops significantly for OMC: at $\epsilon$ 0.25 to 0.32 and at $\epsilon$ 0.1 to 0.13, while it remains high for SMC (0.91 to 0.83). This is the result the inability of every $\boldsymbol{u}_i$ to achieve $\rho < \epsilon$, whereas for SMC, the algorithm allows them to "drop" their random numbers and effectively switch to another. This was verified by running an expensive fine-grained optimization, resulting in 32.9% and 13.6% optimized particles having $\rho$ under $\epsilon$ 0.25/0.1. Taking this inefficiency into account, OMC still requires 130 simulations per effective sample v 165 for SMC (ie 17/0.13 and 136/0.83).

where $r_{u_m} = 1 + \sqrt{2}\,\mathrm{erf}^{-1}(2u_m - 1)$. We put a positive constraint on $\theta$: $p(\theta) = \mathcal{U}(0, 10)$. We used 2 statistics, the mean and variance of $M$ draws from the simulator:

$$s_1(\mathbf{x}) = \frac{1}{M} x_m \qquad\qquad \implies\ f_1(\theta, \boldsymbol{u}) = \theta R(\boldsymbol{u}) \qquad \frac{\partial f_1(\theta, \boldsymbol{u})}{\partial \theta} = R(\boldsymbol{u}) \qquad (19)$$

$$s_2(\mathbf{x}) = \frac{1}{M} \sum_m (x_m - s_1(\mathbf{x}))^2 \implies\ f_2(\theta, \boldsymbol{u}) = \theta^2 V(\boldsymbol{u}) \qquad \frac{\partial f_2(\theta, \boldsymbol{u})}{\partial \theta} = 2\theta V(\boldsymbol{u}) \quad (20)$$

where $V(\boldsymbol{u}) = \sum_m r_{u_m}^2/M - R(\boldsymbol{u})^2$ and $R(\boldsymbol{u}) = \sum_m r_{u_m}/M$; the exact Jacobian is therefore $[R(\boldsymbol{u}), 2\theta V(\boldsymbol{u})]^T$. In our experiments $M = 10$, $s(\mathbf{y}) = [2.7, 12.8]$.

### 4.5 Lotka-Volterra

The simplest Lotka-Volterra model explains predator-prey populations over time, controlled by a set of stochastic differential equations:

$$\frac{dx_1}{dt} = \theta_1 x_1 - \theta_2 x_1 x_2 + r_1 \qquad \frac{dx_2}{dt} = -\theta_2 x_2 - \theta_3 x_1 x_2 + r_2 \qquad (21)$$

where $x_1$ and $x_2$ are the prey and predator population sizes, respectively. Gaussian noise $r \sim \mathcal{N}(0, 10^2)$ is added at each full time-step. Lognormal priors are placed over $\boldsymbol{\theta}$. The simulator runs for $T = 50$ time steps, with constant initial populations of 100 for both prey and predator. There is therefore $P = 2T$ outputs (prey and predator populations concatenated), which we use as the statistics. To run a deterministic simulation, we draw $\boldsymbol{u}_i \sim \pi(\boldsymbol{u})$ where the dimension of $\boldsymbol{u}$ is $P$. Half of the random variables are used for $r_1$ and the other half for $r_2$. In other words, $r_{u_{st}} = 10\sqrt{2}\,\mathrm{erf}^{-1}(2u_{st} - 1)$, where $s \in \{1, 2\}$ for the appropriate population. The Jacobian is a $100 \times 3$ matrix that can be computed using one-sided finite-differences using 3 forward simulations.

### 4.6 Bayesian Inference of the M/G/1 Queue Model

Bayesian inference of the M/G/1 queuing model is challenging, requiring ABC algorithms [4, 8] or sophisticated MCMC-based procedures [18]. Though simple to simulate, the output can be quite

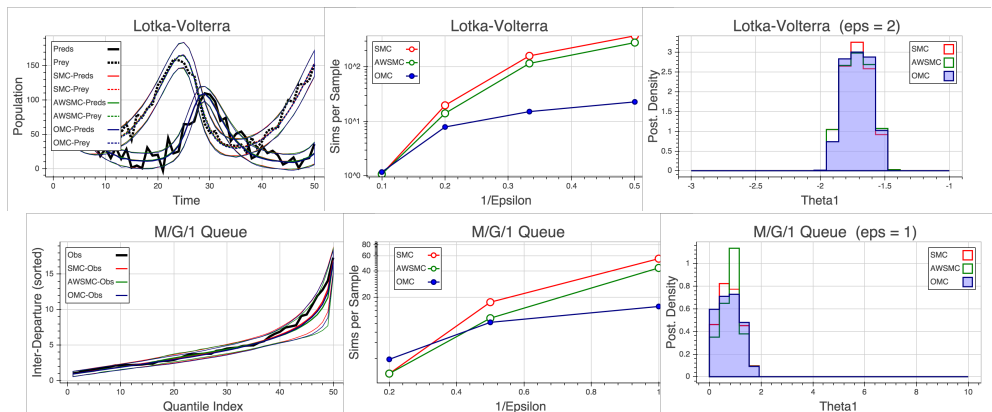

Figure 4: **Top:** Lotka-Volterra. **Bottom:** M/G/1 Queue. The left plots shows the posterior mean $\pm 1$ std errors of the posterior predictive distribution (sorted for M/G/1). Simulations per sample and the posterior of $\theta_1$ are shown in the other plots. For L-V, at $\epsilon$ 3, the SS for OMC were 15 v 116/159 for AW/SMC, and increased at $\epsilon$ 2 to 23 v 279/371. However, the ESS/n was lower for OMC: at $\epsilon$ 3 it was 0.25 and down to 0.1 at $\epsilon$ 2, whereas ESS/n stayed around 0.9 for AW/SMC. This is again due to the optimal discrepancy for some $u$ being greater than $\epsilon$; however, the samples that remain are independent samples. For M/G/1, the results are similar, but the ESS/n is lower than the number of discrepancies satisfying $\epsilon$ 1, 9% v 12%, indicating that the volume of the Jacobians is having a large effect on the variance of the weights. Future work will explore this further.

noisy. In the M/G/1 queuing model, a single server processes arriving customers, which are then served within a random time. Customer $m$ arrives at time $w_m \sim \mathrm{Exp}(\theta_3)$ *after* customer $m-1$, and is served in $s_m \sim \mathcal{U}(\theta_1, \theta_2)$ service time. Both $w_m$ and $s_m$ are unobserved; only the inter-departure times $x_m$ are observed. Following [18], we write the simulation algorithm in terms of arrival times $v_m$. To simplify the updates, we keep track of the departure times $d_m$. Initially, $d_0 = 0$ and $v_0 = 0$, followed by updates for $m \geq 1$:

$$v_m = v_{m-1} + w_m \qquad x_m = s_m + \max(0, v_m - d_{m-1}) \qquad d_m = d_{m-1} + x_m \qquad (22)$$

After trying several optimization procedures, we found the most reliable optimizer was simply a random walk. The random sources in the problem used for $W_m$ (there are $M$) and for $U_m$ (there are $M$), therefore $u$ is dimension $2M$. Typical statistics for this problem are quantiles of $\mathbf{x}$ and/or the minimum and maximum values; in other words, the vector $\mathbf{x}$ is sorted then evenly spaced values for the statistics functions $\boldsymbol{f}$ (we used 3 quantiles). The Jacobian is an $M \times 3$ matrix. In our experiments $\theta^* = [1.0, 5.0, 0.2]$

## 5 Conclusion

We have presented Optimization Monte Carlo, a likelihood-free algorithm that, by controlling the simulator randomness, transforms traditional ABC inference into a set of optimization procedures. By using OMC, scientists can focus attention on finding a useful optimization procedure for their simulator, and then use OMC in parallel to generate samples independently. We have shown that OMC can also be very efficient, though this will depend on the quality of the optimization procedure applied to each problem. In our experiments, the simulators were cheap to run, allowing Jacobian computations using finite differences. We note that for high-dimensional input spaces and expensive simulators, this may be infeasible, solutions include randomized gradient estimates [22] or automatic differentiation (AD) libraries (e.g. [14]). Future work will include incorporating AD, improving efficiency using Sobol numbers (when the size $u$ is known), incorporating Bayesian optimization, adding partial communication between processes, and inference for expensive simulators using gradient-based optimization.

#### Acknowledgments

We thank the anonymous reviewers for the many useful comments that improved this manuscript. MW acknowledges support from Facebook, Google, and Yahoo.

## Footnotes

\*Donald Bren School of Information and Computer Sciences University of California, Irvine, and Canadian Institute for Advanced Research.

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
