[Reviews · NeurIPS 2015]

Submitted by Assigned_Reviewer_1

The ABC procedure discussed in this paper relies on

1- the availability of the mapping allowing to simulate pseudo-observations as a function of random number u (i.e. uniform r.v.) and the availability of its Jacobian.

2- the possibility to solve the optimization problem y=f(theta,u) in theta.

3- Dtheta > Dy

Note that there are quite a few ABC problems where this mapping is not made available to the statistician. Also u could have a random number of components. Finally in complex models, solving the optimization problem y=f(theta,u) could be challenging.

However, in scenarios where 1, 2 and 3 are satisfied, the paper presents an original and interesting scheme.

Quality: good

Clarity: good

Originality: original to the best of my knowledge

Significance: the proposed scheme requires more assumptions on the model than most ABC procedures which might not be satisfied for complex problems where ABC is typically used (e.g. population genetics). It would have been good to explore such an application.
Summary: The paper presents a new original approach to Approximate Bayesian Computation which relies on an optimization procedure. Whenever applicable, this method could be an interesting alternative to standard procedures.

Submitted by Assigned_Reviewer_2

Likelihood-free inference refers to methods used when it is expensive or impossible to evaluate the likelihood function and which instead perform inference by simulating data from the model. Existing methods, such as ABC, simulate data from many parameter values and accept only close matches to the observed data. This is wasteful as most simulations are typically poor matches. This paper proposes Optimisation Monte Carlo (OMC), a more efficient approach which views the simulator as a function f(,u) where  is the unknown parameters and u is a vector of U(0,1) draws. The idea is perform many iterations of the following: draw a u vector, minimise ||y-f(,u)|| (distance between observations and simulator output) to find the best , and weight it according to the the volume of nearby  values providing reasonably good fits. This weight is shown to depend on the Jacobian of f.

OMC is an appealing novel and simple strategy. The paper is clearly written, original and seems technically correct. There are also several examples in which OMC convincingly beats state-of-the-art alternatives. However I have some doubts about significance: the examples are for fairly simple models and I am not convinced the approach will succeed for more complicated scenarios. In particular, I think in all the examples f(,u) is a smooth function of . If the simulator involves branching decisions affected by  this will not necessarily be the case and optimisation could become a lot more difficult, as it may be necessary to search every branch.

A few other points are as follows:

* I'd be interested to know how expensive a typical optimisation step is and whether it affects the runtime per sample of OMC compared to ABC methods. * For some models I imagine only a subset of u space is interesting and it would be nice to have a method which learns and concentrates on this. This is not really a criticism of OMC, as ABC methods don't do this either. * It might be interesting to comment on links with "Efficient likelihood-free Bayesian Computation for household epidemics" by Peter Neal, which seems to share some of the ideas used in OMC.

Finally some possible typos:

* Pg 3, "volume proportional" -> "volume is proportional" * Pg 4 equation (8), is hat needed on first f? * Pg 5, "this problem is" -> "this problem has" * Pg 6, "is simply the" -> "is simply"
Summary: This paper offers an appealing novel strategy for likelihood-free inference. However I have some doubts about whether it will work for more complicated problems, as the optimisation step required may become very difficult.

Submitted by Assigned_Reviewer_3

The paper describes a new approach to Approximate Bayesian Computation whereby computation can be trivially parallellized. The approach relies on having access to an expression relating a vector of random numbers u to the simulated pseudo-data x. Rather than the traditional approach where, for a given theta, pseudo-data x are generated and compared against the observed y, this paper proposes first drawing the random number u, and then choosing theta such that the resulting x are within a distance epsilon of the observed y. The approach is similar to a paper which appeared in the literature recently (as the authors acknowledge), but it is more refined and investigated.

The paper is generally well-written and the approach is interesting. I have a few questions/comments.

* The main drawback I foresee with this method is that, in complicated setups, no theta value within the epsilon-ball will be found, ending up with repeated rejections of the random u's.

* In most typical cases, recovering a multi-dimensional theta is non-trivial, so the general applicability of the algorithm isn't clear.

Minor comments:

* The use of the term 'correctness' is somewhat unusual.

* The authors claim that 'care must be taken so that posterior sampling targets the correct distribution'. Actually there is little guarantee within ABC for targetting the true posterior, and this method won't target the true posterior either.

* U(.) is used without being defined * Lokta should read Lotka * Figure 2, the caption says AW/SMC uses 20 SS - shouldn't there be two numbers there? Is it 20/20?

* Section 4.1: What is M? * A few equations (eg 9) are not punctuated. * Some typos: 'maybe' should read 'may be' (line 132), 'there for' should read 'therefore' (line 414).
Summary: The paper is interesting and well-written. Although many questions remain, the approach is quite novel.

Submitted by Assigned_Reviewer_4

This paper is concerned with an algorithm termed as Optimization Monte Carlo which in other words claims to be an efficient embarrassingly parallel ABC algorithm.

When I first read the abstract, that reminded the coupled-ABC (Neal, Stats and Comp, 2012, vol 6) in which the one supposes that there exists a random vector U such that U is independent of the model parameters  and given u, a realisation of U, a simulated data set x can be constructed as a function of u and  . That is exactly what is made of use in this paper but I think that this paper follows a different path and in particular the OMC step.

However, I think it is a little bit unclear as to what exactly is going on and I have found Section 3 a bit hard to follow.

In particular where does the optimisation and the jacobian come into it.

1. The author(s) should have a look at the paper by Neal and comment on what is different in their algorithm.

2. The example appears convincing and although examples 4.1 and 4.3 are useful, they probably belong to an appendix. What about implementing such an approach on some data from a Lotka-Volterra model which often appears in the ABC literature?

3. Am I right to think that this proposed algorithm will not perform that well when dim(theta) is relatively large?

4. What sort of effect does a failure in the optimisation to the resulting approximate posterior?
Summary: This may be an interesting method to do ABC in parallel and efficiently with an algorithm that is proposed although the idea of breaking down the simulation from the model step into a random uniform number generator and a deterministic function between this function and the parameters is not a new one (see below)

Author Feedback
Author rebuttal: We thank all reviewers for taking their time and providing us with their thoughtful comments. We agree with many of the comments and suggestions brought up in the review. We'd like to help clarify any uncertainties and discuss some of the criticisms.

Neal (coupled ABC)

We were unaware of this work; thanks for pointing it out! After a first read, we are able to make a few limited comments and compare with our work. The most obvious similarity is the reparameterization of the simulator as a deterministic function of (independent) u and theta, ie x = f(theta,u). This allows algorithms to change the sampling order, by first drawing u's from their prior, then theta's are sampled conditional on the u's. We believe this is a powerful approach and opens the door to many new ABC algorithms. Beyond this similarity, it appears to us that the coupled ABC is inherently a sampling algorithm (with key rejection steps and randomized acceptance of non-rejected particles), whereas the main driver of our algorithm is optimization (which, in our case, requires the Jacobian computation for the final particle weighting). We also found the Neal's algorithm constructions are somewhat problem specific. With OMC, the algorithm is general, but has the freedom of plugging in your favorite optimizer. We are not sure experimentally how they compare, but our impression is that it will be less efficient than OMC due to the rejection steps. It would definitely be interesting to compare OMC with coupled-ABC empirically.

Experiments are limited to simple, smooth examples and may not extend to non-smooth f (such as those presented by branching processes).

We definitely agree that the smoothness of f (conditioned on u) could be an issue. We haven't studied branching processes specifically, but we do not believe that they are non-smooth (conditioned on u, which would define the branches of the tree). We could be wrong on this. In any case, the optimization procedure may indeed be difficult, but we don't think this is solely a problem for OMC, since for any sampling algorithm searching over trees would be necessary. OMC would have the advantage of using tree searching optimization algorithms.

Repeated rejections of random u's.

We agree that this will/is a problem for small epsilon and more challenging problems. This occurs in our experiments to some degree (Figure 3). However, this is not just an issue for OMC but for ABC in general, when the epsilon parameter is set too small. Typical ABC algorithms can effectively switch to another set of u at their current (or nearby) theta location (by sampling p(x|theta)). In OMC, we have fixed u. Consider a set of n particles sorted by their distances rho. This list implicitly defines a set of epsilons, such that for any epsilon e, there are n_e <= n particles inside the epsilon ball (ie OMC is an "anytime" algo). One strategy (mentioned in paper) is to allocate resources to the particles with the highest rho, decreasing the largest epsilon. If no progress is made then this particle's optimization has converged and we can either add a new particle (draw u from the prior) and add it to the list, or move up the list to the other particles. There are many possible algorithmic avenues to explore.

OMC in (high)-Dtheta

Our derivation of OMC assumed that Dy >= Dtheta. As long as this is satisfied, we do not think OMC would suffer any differently than regular ABC, though OMC would have the advantage of generating independent samples (and using sophisticated opt. algos), whereas high Dtheta might require random walk type ABC-MCMC (highly correlated samples). Reviewer perceptions may be that for OMC there is a fixed epsilon and a particles outside the e-ball are rejected. This is partly true and is the story presented in the paper because we compared directly with SMC. Another view (outlined above) is that the distances of all particles define a continuum of *satisfied* epsilon values. At *anytime*, the user can select an epsilon from that list that has adequate numbers of particles. In high Dtheta, this view remains the same. The epsilons will be larger, but we don't believe the high-dimensions will affect OMC any more that regular ABC.

Relying on the availability of f(theta,u) (as function of u) and jacobian

We haven't come across a (stochastic) simulator that doesn't allow controlling, at a minimum, the seed of the random number generator inside the simulator (though this could be the case for legacy black-box simulation code). But we agree, if we can't control the randomness, we cannot use OMC. Wrt computing the jacobian, there are challenges, including smoothness of f, high Dtheta etc. We agree these could be issues but think they can be handled in various ways .

Relying on Dtheta > Dy (assuming R7 meant Dtheta <= Dy)

We acknowledge OMC does indeed have this limitation. We are actively working on OMC in this ill-posed scenario.